# Numerical Investigation of Outflow of Non-Metallic Inclusions during Steel Refining in the Ladle

**DOI:** 10.3390/ma15093039

**Published:** 2022-04-22

**Authors:** Piotr Migas, Marta Ślęzak, Mirosław Karbowniczek, Stanisław Szczęch, Andrzej Hornik

**Affiliations:** 1Department of Ferrous Metallurgy, Faculty of Metals Engineering and Industrial Computer Science, AGH—University of Science and Technology, 30 Mickiewicza Av., 30-059 Krakow, Poland; pmigas@agh.edu.pl (P.M.); mkarbow@agh.edu.pl (M.K.); 2Cognor SA HSJ Department in Stalowa Wola, 1 Kwiatkowskiego, 37-450 Stalowa Wola, Poland; sszczech@hsjsa.pl (S.S.); ahornik@hsjsa.pl (A.H.)

**Keywords:** non-metallic inclusions, ladle, numerical simulation

## Abstract

The article presents the results of numerical simulations of liquid steel flow in the main steelmaking ladle. The paper analyses the mechanism of the outflow of non-metallic Al_2_O_3_ and MnS inclusions with diameters in the range of 4–27 µm. The simulations were performed with ANSYS Fluent software. In order to determine the shape and size of non-metallic inclusions formed in the main ladle during steel refining, the collected samples of liquid metal were analysed using a scanning microscope with SEM/EDS and LM (light microscopy). Simulation tests and calculations were carried out for the case of steel refining under the conditions of the Cognor SA HSJ Department in Stalowa Wola (Poland). The presented method of using simulation tests to optimize the technology of steel refining in the ladle is an example. The analysis of the results shows that the gas flow in the metal volume has the greatest impact on the outflow of non-metallic inclusions in the steelmaking ladle.

## 1. Introduction

Currently, steel production is associated with ensuring the best possible cleanliness. The production of steel is therefore divided into two main stages: in the first step, the main goal is to obtain a metal bath as quickly as possible (from an economical point of view); in the second stage, the chemical composition and the required metallurgical purity of the produced alloy are determined. The mentioned second stage, i.e., refining and purification of steel from impurities, is a key moment in the production of iron alloys. Mathematical modelling of phenomena occurring during metallurgical processes is commonly used to understand their complexity. The process of refining iron alloys has been physically and numerically modelled for several years. It is assessed, i.e., the efficiency of the ladle or a tundish, using macro-mixing characteristics that give information about how much time a fraction of the volume of liquid spends in the reactor [1].

Conducting tests of the behaviour of liquid metal under industrial conditions in an electric arc furnace, in a ladle during refining, and in a tundish during casting in a COS machine is difficult; some measurements are impossible. For this reason, such studies are carried out in laboratory conditions using physical (water) models, or with the use of mathematical modelling.

The nature of the liquid metal flow in metallurgical reactors can be described using equations expressing the conservation of momentum for a moving fluid, namely Navier–Stokes equations. Systems of differential equations can be solved with the use of commercial computer programs in the field of Computational Fluid Dynamics (CFD), which are based on commercial and the authors’ calculation codes. Modelling makes it possible to visualize the flow structure, which allows for a better understanding of the analysed issue, as well as the prediction of various variables and optimization of process parameters [1].

When studying fluid movement (multiphase flows) one can use many methods of mathematical description [2]. One method is to use a Lagrange grid to extract infinitely small volume elements from the fluid and track their motion. The equations of motion are solved directly for every single particle. Another solution is to describe a two-phase flow using the Euler approach [3,4]. The method consists of determining the velocity and density of a fluid at any point in space at any moment. In this case, it is necessary to provide the boundary conditions [2,3]. Numerical modelling is commonly used in steel metallurgy to analyse problems such as steel flow [5,6,7,8,9,10,11,12,13,14,15,16,17,18,19,20,21,22] and turbulence phenomena in a tundish [23,24,25], heat and mass exchange in a reactor [26,27], removal of non-metallic inclusions [28,29,30], and improvements in liquid steel casting conditions during main ladle changes. 

As the above analysis shows, there are many publications describing (analysing) the issues of modelling and simulating the behaviour (movement) of metal in the ladle. On the other hand, it is difficult to find in the literature information on the behaviour (flow) of non-metallic inclusions from the liquid metal during the refining of steel in the ladle.

Liquid steel refining is one of the main tasks of steel treatment to decrease the level and remove non-metallic inclusions (floating to the slag). The refining process is mainly deoxidation. One of the basic conditions for the production of high-quality steels is the use of technologies that enable impurities precipitated in the liquid state, so-called non-metallic inclusions, to be controlled and to maximize their removal. This process is carried out during the refining of the steel, e.g., in the ladle (LF step), and also partly during the casting process [14,15,16]. Non-metallic inclusions are indispensable elements of the steelmaking process. Their chemical composition depends on the presence of certain elements in the liquid steel grades and the type of alloying additives and deoxidizers introduced. The removal of impurities from the liquid metal bath takes place in the ladle and the mould of the continuous casting machine, with the ladle refining processes playing a major role. The effectiveness of the operation of removing non-metallic inclusions from the liquid steel is determined by a number of factors, such as steel melting point, chemical composition and physical forms of the precipitates, particle wettability by liquid steel, movements and flows of liquid and its mixing parameters, argon flows, as well as the dimensions and geometry of the considered metallurgical units [17]. Non-metallic inclusions occurring in steel can be divided according to their characteristics, that is, origin, type, size, shape, chemical composition, or the state of the inclusions in the liquid steel. When divided by origin, two types of inclusions are distinguished: endogenous inclusions (formed as a result of chemical reactions within the liquid), and exogenous inclusions (coming from outside the liquid). From the point of view of the type of inclusions, the most important are oxides (Al_2_O_3_, SiO_2_, FeO, MnO), sulphides (FeS, MnS), and nitrides (AlN, CrN); solutions of sulphides and oxides of the general formula (Mn, Fe) S-FeO can also be formed [31,32,33,34]. Another category in the inclusion morphology includes their sizes and shapes. Inclusions of a globular or irregular (sometimes highly developed) shape may be encountered. All these parameters are also dependent on the chemical composition and state of aggregation and significantly influence the behaviour of inclusions in the liquid metal bath. The term “behaviour” of inclusions should be understood as their flotation, coagulation, sedimentation, and assimilation to the slag surface, or “sticking” to the walls of the steelmaking unit. This “behaviour” depends on the intensity of mixing of the liquid metal and slag. A precise mathematical description of the phenomena occurring during the formation, movement, and outflow of non-metallic inclusions in liquid metal is a very complex task. Practical implementation of such simulations for a specific steelmaking unit is currently possible with the use of specialized, commercial computer programs. The formation of the non-metallic phase composed of liquid/solid non-metallic inclusions favours the process of their removal from the metal bath, which is due to the agglomeration mechanism, i.e., their joining as a result of collisions, which in turn facilitate their outflow from the liquid solution and their assimilation to the slag. The surface phenomena and the wettability of non-metallic phase particles by liquid steel and the technological parameters of the steel refining stage also play significant roles in this process. The ability to assimilate and dissolve inclusions in the slag is an important parameter determining the effectiveness of removing non-metallic inclusions during the refining of a metal bath. The particle size determines the outflow conditions and, consequently, the possibility of their removal. Since the non-metallic phase in liquid steel is diversified in terms of particle size (from a few nanometres to several hundred micrometres), an additional factor determining its removal is the increase due to the diffusion and dynamic mechanism (agglomeration), with the latter playing a dominant role in the processes taking place in the tundish [18,24].

The production of high-quality steel, especially of high metallurgical purity, is conditioned by the minimization of the amount of non-metallic inclusions. The problem of their effective removal from steel in the production process, from the technological point of view, includes the control of their formation and behaviour in the liquid metal and slag in the ladle during the refining period and in the tundish and the mould during continuous casting, as well as during crystallization. The inclusions formed during solidification of the steel remain in its volume, and it is not possible to remove them. The effectiveness of the operation of removing non-metallic inclusions from the liquid steel is determined by a number of factors, such as: chemical composition systems, melting point and physical form of non-metallic precipitates, wettability of particles (liquids) by liquid steel, the ability of the slag to assimilate inclusions, the movement of liquid steel and its mixing parameters, as well as the dimensions and geometry of the metallurgical units under consideration, mainly the ladle.

The article presents the results of numerical simulations of the outflow of two types of non-metallic inclusions, Al_2_O_3_ and MnS, in the most common diameters of 4, 12, 18, and 27 µm during the movement of liquid steel (refining) in the ladle for two grades: 18CrNiMo7-642 and CrMo4. The proposed methodology of analysis is an innovative approach to the issues of numerical simulation of the steel refining process in terms of removing non-metallic inclusions (improvement of metallurgical purity). The simulations were performed with ANSYS Fluent software. Simulation tests and calculations were carried out for the case of steel refining under the conditions of Cognor SA HSJ Department in Stalowa Wola (Poland). The obtained test results and their analysis enable the optimization of the optimal technological parameters of the refining process in industrial conditions. The presented method of using simulation tests to optimize the technology of steel refining in the main ladle is the example. Based on the simulation results, technological guidelines were formed and implemented in the industrial conditions of Cognor SA.

## 2. Materials and Methods

The model and numerical simulations of behaviour of the liquid metal and non-metallic inclusions in the steelmaking ladle were developed for refining two grades of steel with different carbon contents.

Selected simulation parameters, namely the shape and geometry of the ladle, the location of the gas (argon) injection fitting, the argon flow rate, as well as the steel grades and the refining temperature of the liquid metal bath are appropriate, and they were applied in the industrial process conditions of the Cognor SA HSJ Department in Stalowa Wola. The chemical compositions of the analysed steel grades are presented in Table 1.

The densities of analysed steel grades were determined from the relationship, taking into account the temperature and carbon content [34]:ρ_st_ = (8319.49 − 0.835·T)/(1 − 0.01·%C),(1)
where
ρ_st_—density of liquid steel, kg/m^3^,T—temperature—means the temperature during the refining of steel in the ladle; in the production process it was 1620 °C for 18CrNiMo7-6 steel and 1610 °C for 42CrMo4 steel,%C—carbon content in steel, in wt%.


The results of the calculations of the melting point and steel density are presented in Table 2.

Based on data from [31], the viscosity of the liquid steel was assumed to be 0.0067 Pas. Generally, for construction of simulation models of industrial processes with significant turbulent flows, three basic approaches are used: DNS—Direct Numerical Simulation, LES—Large Eddy Simulation, and RANS—Reynolds Averaged Navier–Stokes Simulation [1,2,3]. The RANS variant is the most frequently used approach in solving the problems of modelling turbulence flows. It is characterised by the ability to solve the N–S equations in a relatively short time; many different flow models are available, and most turbulent movements can be modelled. The approach to solving the problem in the RANS variant is characterized by a system of three equations for plane problems (2D) and four equations for spatial problems (3D). For this variant, the standard k–ε model [2] is available in ANSYS Fluent. The k–ε turbulence model belongs to the group of models in which, for modelling the components of the stress tensor, the Boussinesq’a hypothesis is used, according to which turbulent (Reynolds) stresses behave similarly to other stresses in the fluid and can be described as proportional to the deformation speed [35,36]. The standard model, k–ε, is written by two general equations (k—the so-called kinetic energy of turbulence, and ε—energy dissipation). At a high Reynolds number flow, the k–ε model used is characterised by the stability of the system of turbulence equations, and convergence to a steady state from any set of initial conditions [36].

In this case, numerical simulations of argon blowing of liquid steel were carried out using the commercial software ANSYS Fluent (Ansys, Inc.,Canonsburg, PA, USA). Computational mesh was built up in the ANSYS pre-processor software and contained 324 371 control volumes. Simulations were performed for a gas blowing intensity of 25–50 $\frac{{\mathrm{dm}}^{3}}{\min}$. The Discrete Phase Model (DPM) was used to model argon bubbles with a uniform diameter distribution of 0.015 m. In current simulation for the side wall and bottom of the ladle, the stationary wall boundary condition (u = v = w = 0) was applied. Additionally, it was assumed that the steel level is a flat free surface—a stationary wall with zero shear stresses (τ_xy_ = τ_xz_ = τ_yz_ = 0)—and the presence of slag protecting the metal was omitted. Argon bubbles flow out from the porous plug and float in the model fluid. Gas bubbles leave the system after reaching the liquid-free surface. Contact with other walls causes their reflection. For analysis, the Standard k–ε model was used to describe turbulence. Discretization equations were derived from the governing equations and were solved by using an implicit finite difference procedure called the simplec algorithmic. A schematic view of the ladle with boundary conditions applied for the numerical model is shown in Figure 1 and Figure 2. The dimensions of the ladle are as follows: diameter at the bottom 1932 mm, diameter at the top 2058 mm, height 2770 mm. The ladle has one gas-permeable fitting located in the bottom, at a distance of 440 mm from the axis.

Table 3 presents assumed designations and parameters of the simulation variants.

Subsequently, research on the behaviour (flotation) of non-metallic inclusions at the time of refining of steel in the ladle was carried out. The same geometrical parameters of the ladle, the system of the gas-permeable piece, as well as the physicochemical parameters for the same two grades of steel were assumed in the tests. For the analysis of the behaviour of non-metallic inclusions, it was assumed that they appear uniformly over the bottom of the ladle. The time of removing inclusions from the ladle was analysed. The trajectories of the particles were plotted on the calculated flow field of the liquid steel, which means that the particles follow the movement of the liquid steel. For simulation purposes, the type, size, and shape of the particles (non-metallic inclusions) had to be assumed. In order to determine these data (parameters), tests were performed, first on the microstructure, and then on the macrostructure of samples taken from the ladle under industrial conditions during the steel refining process. In order to identify the shape and size of non-metallic inclusions formed in the main ladle during steel refining, samples of liquid metal were taken and quickly quenched. The samples were examined using the FEI Versa 3D scanning electron microscope (FEI, Hillsboro, OR, USA). EDS microstructural analysis was performed on polished samples using a voltage of 20 kV and a spot size of 3. Sample photos and analyses of the chemical composition of non-metallic inclusions were made for each melt. The observed inclusions were mainly fine, spherical manganese sulphides or spherical fine aluminium oxides with a very complex chemical composition. Elongated manganese sulphides dominated in samples taken from the final products. Small spherical manganese sulphides were also occasionally present. In many analyses it was shown that the sulphides nucleated on the oxides. The observed fine oxides were mainly aluminium oxides. Aluminium oxides with an admixture of calcium, magnesium, and silicon were also observed, occasionally with an addition of chrome or titanium. The presence of a few nitrides was found.

Then, quasi-quantitative analyses of the macrostructure content of non-metallic inclusions in the samples were performed. The analyses were performed using a Nikon Elipse light microscope (Nikon Corporation, Shinagawa Intercity Tower C, Konan, Minato-ku, Tokyo, Japan). The research was carried out on non-etched specimens. A series of 23 photos was taken at the same magnification in different areas of the samples (in total, an area of analysis of approx. 20 mm^2^), and then a calculation and evaluation of inclusions were made on the basis of the morphology and colour of the non-metallic inclusion. Grey, longitudinal, and approximately spherical inclusions were classified as manganese sulphides, approximately spherical black inclusions were classified as oxides, and gold-coloured inclusions were classified as nitrides. Due to the resolution of the method, spherical inclusions smaller than 3 µm, which could be oxide–sulphide conglomerates, were counted as oxides. After identifying the insertion, the program automatically assigned the longest insertion dimension and described it with a circle. The volume fraction of the inclusions was calculated based on the areas of the circles described in the inclusions. Hence, the determined volumetric fraction should be treated as a comparative between individual samples and not as the actual fraction of non-metallic inclusions in the tested steel samples. The greatest measurement error occurred with very long manganese sulphides. In the case of the oxides and nitrides, due to their morphology, the measurement of their volume fraction could be treated as approximately realistic.

Examples of macrostructure views for 18CrNiMo7-6 and 42CrMo4 grades are shown in Figure 3 and Figure 4, respectively.

The results of the quantitative distribution of inclusions, calculated as a percentage of the total area occupied by inclusions of a given type to the total area of the analysed samples, are presented in Table 4 and Table 5.

As regards results from the analysis of the images (Figure 3 and Figure 4), the greatest number of non-metallic oxide-type inclusions were identified, including mainly Al_2_O_3_ for 18CrNiMo7-6 and 42CrMo4 grades (0.22% on average for both grades) and sulphides, and mainly MnS for steel in the 42CrMo4 grade (0.88% on average).

At the same time, based on the measurements of the dimensions of the inclusions, it was noticed that inclusions of various shapes appeared, but it can be simplified to assume that their sizes were within the range from <3 µm to more than 27 µm. At the same time, it was simplified that all inclusions had a spherical shape.

The planned simulations of the outflow of non-metallic inclusions during the refining of steel in the ladle should enable the development of technological guidelines for the production of steel with higher metallurgical purity. New products are expected to meet increased requirements concerning the non-metallic inclusions according to the ASTM E45-13 (method A), namely type B (thin—0.0/thick—0.0), type C (thin—0.0/thick—0.0), type D (thin—0.5/thick—0.5), and the ISO 4967:2013 (method A) for type D (0.5/0.5, DS 0.5).

Taking into account the results of the conducted research, the following variants were adopted for the simulation calculations:two grades of steel: 18CrNiMo7-6 and 42CrMo4,two types of the most common non-metallic inclusions: Al_2_O_3_ and MnS,argon flow rate: 25 and 50 L/min,the most common sizes: spherical with diameters of 4, 12, 18, and 27 µm.

The adopted designations and parameters of the simulation variants of the behaviour of non-metallic inclusions are presented in Table 6 and Table 7.

## 3. Results

First, simulations of the metal movement in the ladle by argon stirring were performed. The simulation results were presented in the form of contour maps of the speed of movement in the volume of liquid steel in the refining ladle. Simulations were performed for two values of steel density: 6979 kg/m^3^ (18CrNiMo7-6 grade), and 7004 kg/m^3^ (42CrMo4 grade). There were no significant differences between the analysed densities observed. They were quite similar for both values of the density for the same argon flux. However, changes of the speed fields for different argon flows were found. Exemplary simulation results for the selected steel grade 42CrMo4 and two different flows are shown in Figure 5 and Figure 6.

As can be seen from the velocity distribution of liquid steel in selected control planes and marked metal movement streams in the volume of the ladle during steel refining, the model of metal behaviour was not homogeneous. There were areas of high velocity of the stream of moving metal (red colour), especially in the area of the outflow of the gas stream; there were also zones with very low speeds of movement, close to zero (blue). Zones with no metal movement are called dead zones.

For steel of the 42CrMo4 grade (Figure 5 and Figure 6), dead zones appeared especially in the upper part of the ladle, on the opposite side of the gas fitting for the gas stream flow of 25 L/min, and especially in the lower part of the ladle, on the opposite side of the gas fitting for the flow gas stream of 50 L/min. The dead zones were smaller for greater gas flow.

Then, simulations of the non-metallic inclusions movement in liquid steel in the ladle by argon stirring were performed. The results of simulation calculations for individual variants are shown in Figure 7, Figure 8, Figure 9, Figure 10, Figure 11, Figure 12, Figure 13 and Figure 14. The times (90 s and 180 s) shown in the figures represent the time that elapsed since the particle appeared. The drawings show the trajectories of movement for particles of a given diameter. Particle trajectories were plotted on the calculated liquid steel flow field—the particles followed the movement of the liquid steel.

Figure 7, Figure 8, Figure 9 and Figure 10 for the 18CrNiMo7-6 grade and Figure 11, Figure 12, Figure 13 and Figure 14 for the 42CrMo4 grade show exemplary trajectories for particles of a given diameter. The red lines reflect the trajectories of the non-metallic inclusions (in the number of 340) in the liquid steel after their separation from the solution near the bottom of the ladle. This is followed by the commencement of the movement being the vector sum of the component related to the gravity and the component resulting from the turbulent movement of the metal caused by the blown inert gas (argon). Attention should be paid to the stochastic movement of individual inclusion particles, including their upward movement (outflow), although evidently, their flow is turbulent. The force due to gravity is the effect of the difference between the density of the particle and the molten steel density. This difference in density was approximately double. The turbulence of movement is related to the intensity of the blown inert gas.

As seen from the analysis (Figure 7) of the movement of solid particles (Al_2_O_3_) in the liquid metal, initially the particles began to rise upwards, but with a trajectory at a slight angle from the horizontal, towards the ladle axis. It should be recalled that the simulations assumed the appearance of particles evenly distributed at the bottom of the ladle. After 90 s, one could observe the accumulation of particles in the horizontal plane at a distance of about 10% of the height of the ladle and individual particles that flowed higher. At the same time, a clear stream of particles was visible flowing upwards vertically above the inlet of the argon stream. After 180 s, one could observe the accumulation of particles in the horizontal plane at a distance slightly higher than before and many more particles floating higher, some of which managed to flow up to the upper surface of the liquid metal. At the same time, there is a distinctly increased stream of particles flowing vertically upwards over the argon stream inlet and on the opposite side of the ladle. The described outflow phenomena are of a very similar nature for all analysed sizes of non-metallic inclusions.

As seen from the analysis (Figure 8) of the movement of solid particles (MnS) in the liquid metal, initially the particles began to rise upwards, but with a trajectory at a slight angle from the horizontal, towards the ladle axis, very similarly to Al_2_O_3_ inclusions. Similarly, after 90 s, one could observe the accumulation of particles in the horizontal plane at a distance of about 10% of the height of the ladle, and individual particles that floated higher. In addition, after 180 s, there was an accumulation of particles in the horizontal plane at a slightly higher distance, and many more particles flowing higher. There was also an increased stream of particles flowing vertically upwards over the argon inlet and on the opposite side of the ladle. These phenomena were of a very similar nature for all analysed sizes of non-metallic inclusions

As seen from the analysis (Figure 9) of the movement of solid particles (Al_2_O_3_) in the liquid metal, initially the particles started to float upwards, but with a trajectory of movement at a slight angle from the horizontal, towards the ladle axis. After 90 s, one could observe the accumulation of particles in the horizontal plane at a distance of about 15% of the height of the ladle, which were more concentrated than in the case of the argon flow of 25 L/min, and some individual particles flowed higher. After 180 s, a distinct wide stream of particles could be observed flowing almost vertically upwards, sloping obliquely from the bottom on the opposite side of the gas stream inlet, towards the argon stream inlet. The described phenomena of outflow had a very similar character for all analysed sizes of non-metallic inclusions.

As seen from the analysis (Figure 10) of the movement of solid particles (MnS) in the liquid metal, initially the particles started to float upwards, but with a trajectory of movement at a slight angle from the horizontal, towards the axis of the ladle, as in the case of Al_2_O_3_ inclusions. After 90 s, one could observe the accumulation of particles in the horizontal plane at a distance of about 15% of the ladle’s height. They are more concentrated than in the case of the argon flow of 25 L/min, and individual particles that flowed higher, as was the case for Al_2_O_3_ particles. In addition, after 180 s, a distinct wide stream of particles was observed flowing almost vertically upward, sloping obliquely from the bottom on the opposite side of the gas stream inlet towards the argon stream inlet. These phenomena were of a very similar nature for all analysed sizes of non-metallic inclusions.

All the simulation results obtained for the 42CrMo4 steel grade (Figure 11, Figure 12, Figure 13 and Figure 14) were very similar in terms of quality to the 18CrNiMo7-6 steel grade. The particle trajectories resulting from simulation calculations after 90 s and 180 s, both for Al_2_O_3_ and MnS inclusions, with all analysed dimensions, were practically the same, qualitatively.

Based on the simulation analyses presented above, charts were developed showing the share of removed non-metallic inclusions in time. Figure 15, Figure 16, Figure 17, Figure 18, Figure 19, Figure 20, Figure 21 and Figure 22 show the share of removed non-metallic inclusions for all analysed (above) cases. The proportion of inclusions (in the number of 2200) removed was the percentage of inclusions that were “absorbed” into the slag after a certain time.

The data presented in Figure 15 show the speed of removal of inclusions (particles) from the liquid metal. For Al_2_O_3_ inclusions, about 10% removal was visible after 5 min of refining, for all particle sizes. Half of the inclusions were removed after 12–13 min. On the other hand, over 90% of inclusions were removed after 25–30 min, depending on the size of the particles. Inclusion removal rates for different particle sizes were very similar; the differences between the time needed to remove the assumed amount of inclusions did not exceed one minute in the range of up to 15 min of refining and 5 min in the range of 15–25 min of refining. As can be seen, an important range of removal of inclusions was above 70%, and then, for example, for removing 80% of inclusions, only 20 min was needed for inclusions of class d = 18 µm, while for inclusions of class d = 27 µm, 25 min of refining was needed.

The data presented in Figure 16 show that the rate of removal of MnS inclusions, of all analysed sizes, was the same up to approx. 13 min of refining, and after this time approx. 50% of inclusions are removed. The phenomenon was very similar to that for Al_2_O_3_ inclusions. Slightly different rates were obtained in the refining time range of 13 to 20 min. In contrast, more than 90% of inclusions were removed after 26–28 min, depending on the size of the particles.

The data presented in Figure 17 show that for the argon flow rate of 50 L/min for Al_2_O_3_ inclusions, about 10% of the inclusions were removed after 4 min of refining, for all particle sizes. Half of the inclusions were removed after 8–9 min. On the other hand, over 90% of inclusions were removed after 16–18 min, depending on the particle size. Inclusion removal rates for different particle sizes were very similar, and the differences between the time needed to remove the assumed level of inclusions did not exceed one minute in the range of up to 15 min of refining and 4 min in the range of 15–18 min of refining. As can be seen, an important range of removal of inclusions was the level above 70%; thus, for example, to remove 80% of inclusions, only 13 min are needed for inclusions of class d = 27 µm, while for inclusions of class d = 18 µm, it takes 15 min of refining.

The data presented in Figure 18 show that the rate of removal of MnS inclusions, of all analysed sizes, was the same. Within approx. 4 min of refining, approx. 10% of inclusions were removed. The phenomenon was very similar to that for Al_2_O_3_ inclusions. Slightly different rates were obtained in the refining time range of 5 to 25 min. On the other hand, over 90% of inclusions were removed after 16–18 min, depending on the particle size.

The results of the rate of removed inclusions (both for Al_2_O_3_ and MnS) were very similar in terms of quality for steel of the 42CrMo4 grade.

## 4. Discussion

As it arises from the conducted simulations of the metal movement in the ladle by argon stirring, a change in the shape of the gas stream (along the height of the liquid metal) depends on the argon flow rate; for a flow of 25 L/min—the gas stream deviates from the axis towards the ladle wall, causing the formation of another zone (“dead zone”) with lower rates of steel flow than for argon flow of 50 L/min. At the same time, for a flow of 25 L/min, steel at the level of the stream deflection “slows down”, creating a dead zone, while for a flow of 50 L/min, a ring-shaped zone of slower steel flow is visible. This behaviour and the difference in the steel flow rates at different levels of metal in the ladle are visible for both analysed steel grades. For an argon flow of 50 L/min, a larger area of the slower flow of steel is apparent, but at a lower metal level in the vessel, unlike the argon flow of 25 L/min. The simulations carried out confirm that the argon flow rate has a significant impact on the shape of the gas stream in the metal, and this in turn influences the change of the fields and gradients of the metal movement velocity, both “along the height”, but also “along the radius” of the ladle.

The simulation results of the non-metallic inclusions movement in liquid steel by argon stirring show that the type of non-metallic inclusions (Al_2_O_3_, MnS) as well as their sizes have a small influence on their movement, and trajectory behaviour in the refined liquid metal in the ladle. This phenomenon is probably due to the slight difference in density between Al_2_O_3_ inclusions (3950 kg/m^3^) and MnS (3900 kg/m^3^), i.e., 1.3%. That said, under the temperature conditions of the steel refining process in the ladle, the analysed inclusions are probably in different physical states, namely Al_2_O_3_ in the solid state and MnS in the liquid state, but as shown by the simulation results, it does not have much importance on the trajectories of their motion. This is probably due to the adoption of the same spherical shape of inclusions.

Similarly, the trajectories of particle motion are not affected by the steel grade and the refining temperature. For the analysed steels, the carbon content is different (0.18 and 0.42%), but the liquid density is of greater importance to the phenomena of particle motion. Both analysed steel grades have slightly different densities (6979 kg/m^3^ for 18CrNiMo7-6 steel and 7004 kg/m^3^ for 42CrMo4 steel, i.e., the difference is 0.4%).

However, a significant difference can be noticed as a result of the simulation for two different gas (argon) flow rates of the medium that sets the liquid metal in motion. Particle (inclusions) trajectories for the flow rate of 25 L/min have the character of two practically vertical jets, one above the gas inlet and the other on the opposite side of the ladle. On the other hand, for the flow rate of 50 L/min, the trajectories of movement have the character of one wider stream directed slightly diagonally from the bottom, on the opposite side of the inlet of the gas stream, towards the inlet of the argon stream. The same nature of the trajectory and the movement of particles is observed for all other analysed parameters: steel grade, type and size of particles, and temperature.

The results of share of removed non-metallic inclusions in time show that the type of the analysed non-metallic inclusions, as well as their sizes, have a slight influence on their behaviour (movements, trajectories) in ladle-refined liquid steel. Similarly, the grade of steel (content of carbon and other components) and the temperature of refining have little effect on the behaviour of the non-metallic particles in it.

The results of the rate of removed inclusions during the refining of steel in the ladle confirmed the conclusions resulting from the trajectory analysis of particle movement, including the greatest impact of the refining gas flow rate.

The results of the analyses of the trajectory of the movement of the metal bath in the ladle during refining, especially the rate of outflow of non-metallic inclusions, are very useful for the development of steel refining technology. Knowing the rate of removal of inclusions for specific process conditions, it is possible to improve the design of the refining technique, i.e., to match the refining time to the expected degree of steel purification. It should also be taken into account that the removal of non-metallic inclusions from the metal bath is only one of the elements of the technique process of steel refining in the ladle. During refining, the metal bath must be deoxygenated, its chemical composition must be supplemented, and the appropriate pouring temperature must be obtained, with the ladle requiring to be transferred to the casting machine at the right time, ensuring the sequence of the casting process.

## 5. Conclusions

The developed model and the simulation calculations carried out make it possible to evaluate the behaviour of the liquid metal in the main ladle and in the tundish. The vector field of the velocity of liquid metal movement was assessed, and the trajectories of solid particles’ movement in the volume of liquid steel were simulated for various technological parameters of the steelmaking process. The process parameters were temperature, chemical composition of steel, argon flow in the main ladle, metal flow in the tundish, and the assumed geometrical dimensions of both ladles. Two types of non-metallic inclusions (Al_2_O_3_ and MnS) and four diameters were assumed.The distribution of the velocity of the liquid steel in the selected control planes and the marked fluxes of metal movement in the volume of the main ladle indicate that the model of metal behaviour is not homogeneous. There are high velocity regions of the moving metal stream, particularly in the region of the outgoing gas stream; there are also zones with very low speeds of movement, close to zero (the so-called dead zones).The trajectories of the considered particles in the main ladle indicate that the gas flow in the volume of the metal has the greatest impact on the outflow of non-metallic inclusions. For the flow of 25 L/min, a greater number of circulating inclusions in the volume of the liquid metal is visible, while for the flow of 50 L/min, the inclusions are raised up and assimilate in the upper surface (in the slag). For both flows, it can be seen that there are no major differences in the behaviour of non-metallic inclusions for particles in the range of 4–18 µm, but there are some differences in trajectories for larger particles with a diameter of 27 µm. The above types of behaviour are very similar for both steel grades, both types of inclusions, and also for both temperatures.The gas flow in the volume of the metal has the greatest influence on the outflow of non-metallic inclusions in the main ladle. Regardless of the other process parameters, for a flow of 25 L/min, 50% of the original amount of inclusions was removed after 13–14 min, and 95% of the original amount of inclusions was removed after about 34 min. In addition, independently of the rest of the process parameters, for a flow of 50 L/min, 50% of the original amount of inclusions was removed after 8–9 min, and 95% of the original amount of inclusions was removed after about 21 min.

## Figures and Tables

**Figure 1 materials-15-03039-f001:**
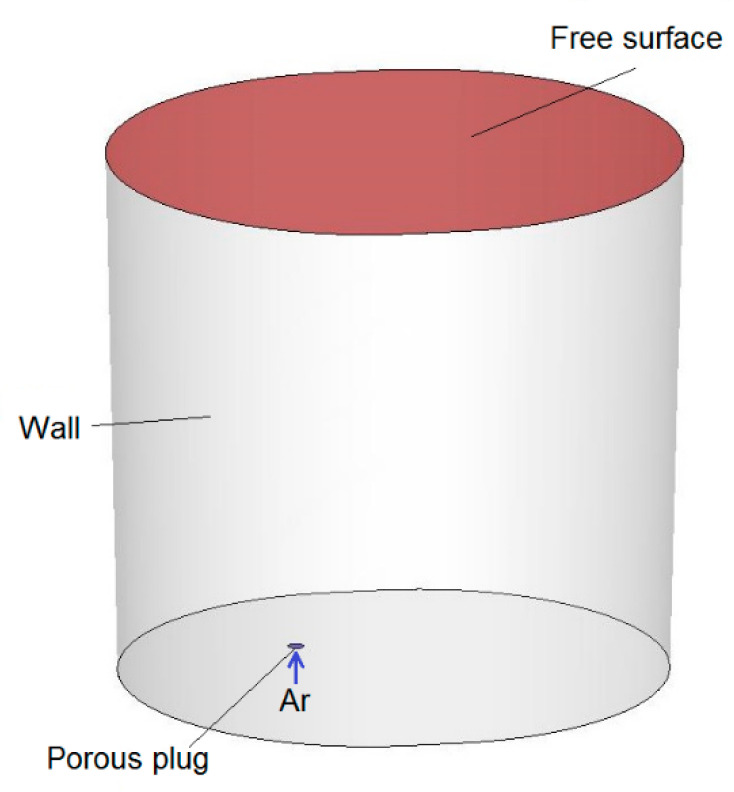
Schematic view of ladle with applied boundary conditions.

**Figure 2 materials-15-03039-f002:**
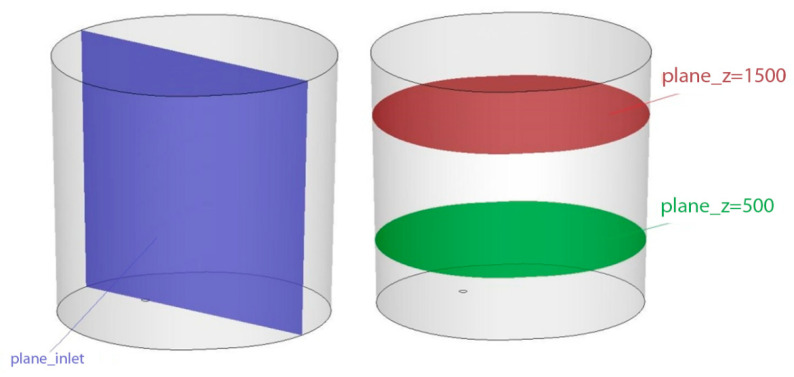
Control planes in the ladle to simulate the movement of the metal bath.

**Figure 3 materials-15-03039-f003:**
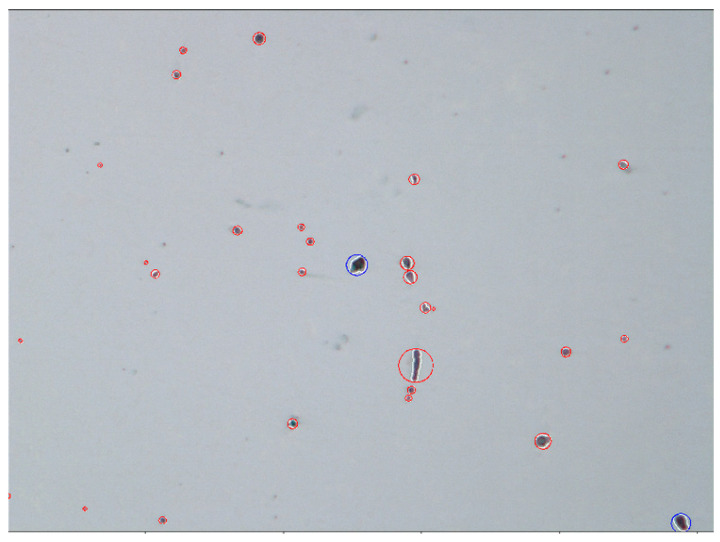
View of the macrostructure of 18CrNiMo7-6 steel with the types of inclusions marked: red colour—oxides, blue colour—sulphides (total dimension of the analysis area 20 mm^2^; magnitude 200×).

**Figure 4 materials-15-03039-f004:**
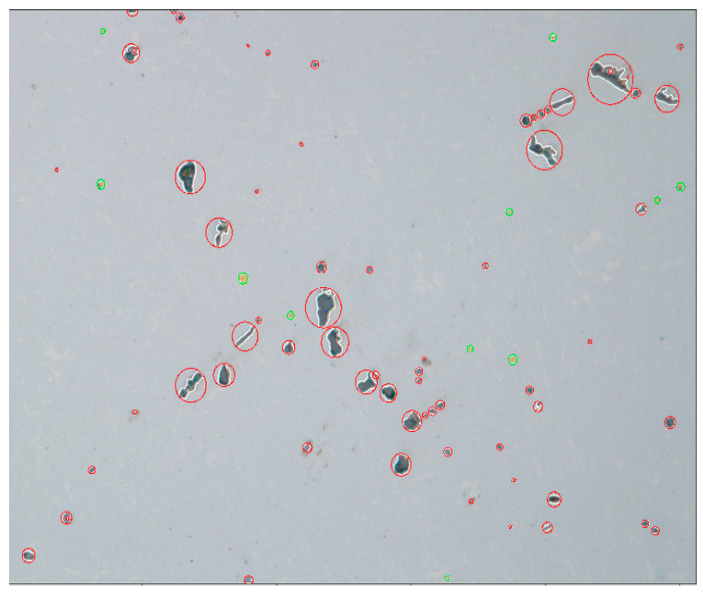
View of the macrostructure of 42CrMo4 steel with the types of inclusions marked: red colour—oxides, green colour—sulphides (total dimension of the analysis area 20 mm^2^; magnitude 200×).

**Figure 5 materials-15-03039-f005:**
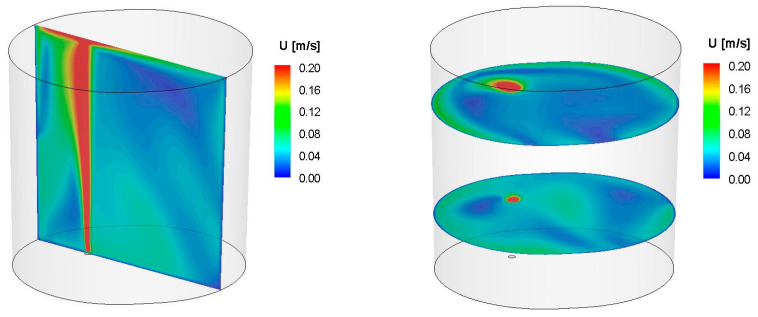
Contour maps of liquid steel velocity in selected control planes for simulation Sim_3 (density = 7004 $\frac{\mathrm{kg}}{m^{3}}$, Ar flow = 25 L/min).

**Figure 6 materials-15-03039-f006:**
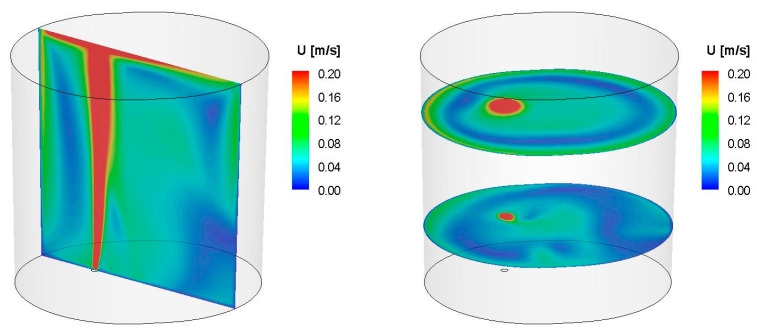
Contour maps of liquid steel velocity in selected control planes for simulation Sim_4 (density = 7004 $\frac{\mathrm{kg}}{m^{3}}$, Ar flow = 50 L/min).

**Figure 7 materials-15-03039-f007:**
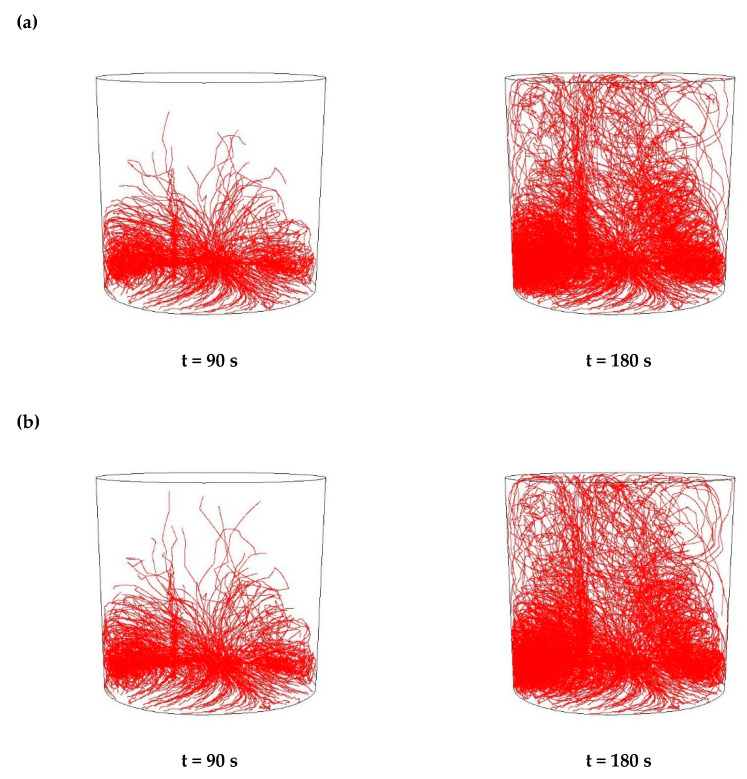
Trajectories of non-metallic Al_2_O_3_ inclusions for the case of argon flow of 25 L/min for the particle diameters of (**a**) 18 µm and (**b**) 27 µm in the 18CrNiMo7-6 steel.

**Figure 8 materials-15-03039-f008:**
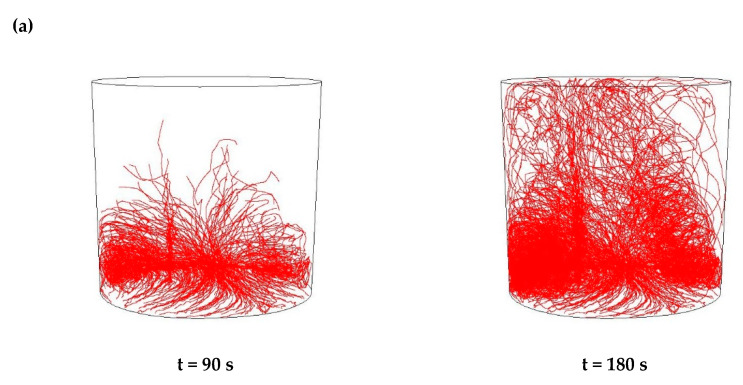
Trajectories of non-metallic MnS inclusions for the case of argon flow of 25 L/min for the particle diameters of (**a**) 18 µm (**b**) 27 µm in the 18CrNiMo7-6 steel.

**Figure 9 materials-15-03039-f009:**
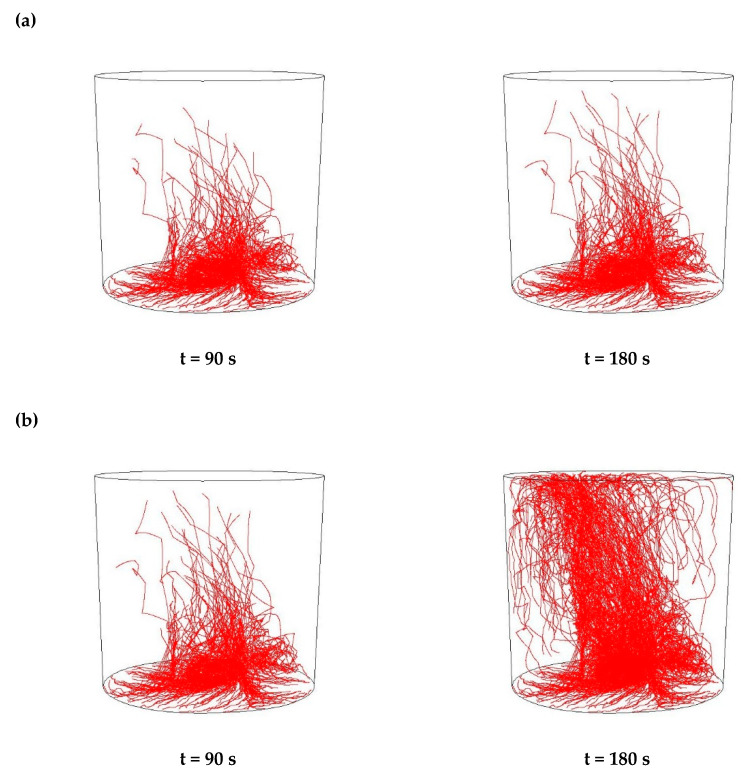
Trajectories of non-metallic Al_2_O_3_ inclusions for the case of argon flow of 50 L/min for the particle diameters of (**a**) 18 µm and (**b**) 27 µm in the 18CrNiMo7-6 steel.

**Figure 10 materials-15-03039-f010:**
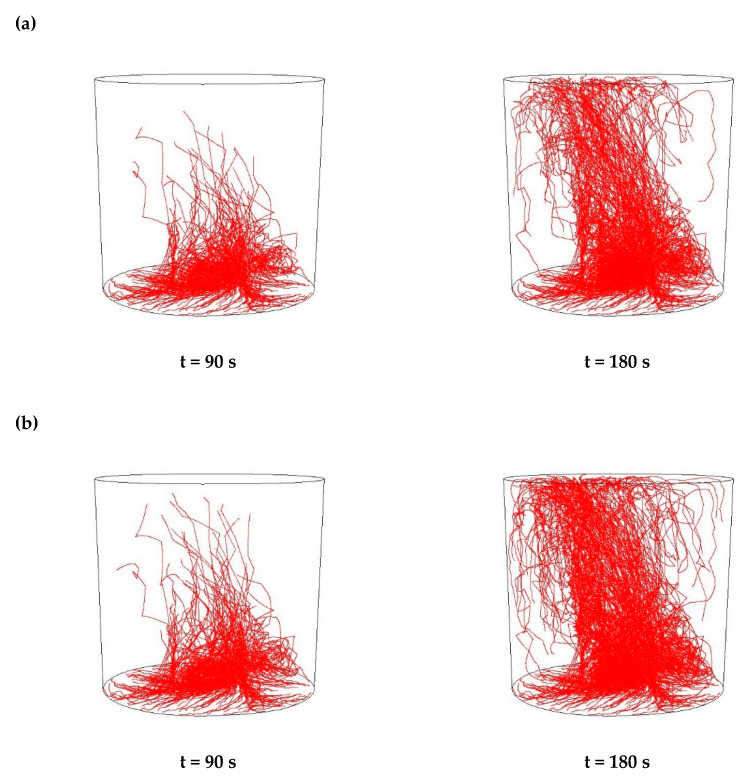
Trajectories of non-metallic MnS inclusions for the case of argon flow of 50 L/min for the particle diameters of (**a**) 18 µm and (**b**) 27 µm in the 18CrNiMo7-6 steel.

**Figure 11 materials-15-03039-f011:**
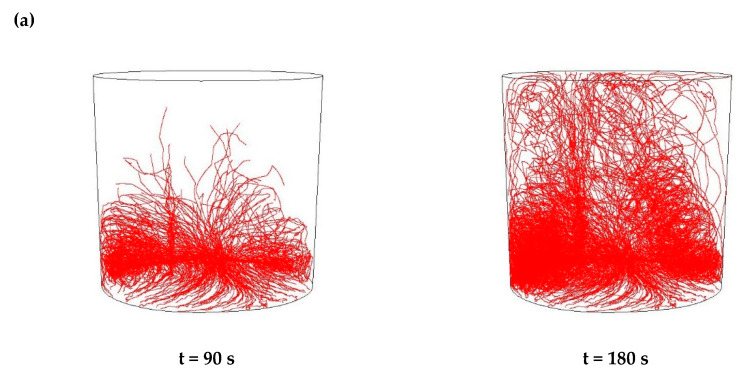
Trajectories of non-metallic Al_2_O_3_ inclusions for the case of argon flow of 25 L/min for the particle diameters of (**a**) 18 µm and (**b**) 27 µm in the 42CrMo4 steel.

**Figure 12 materials-15-03039-f012:**
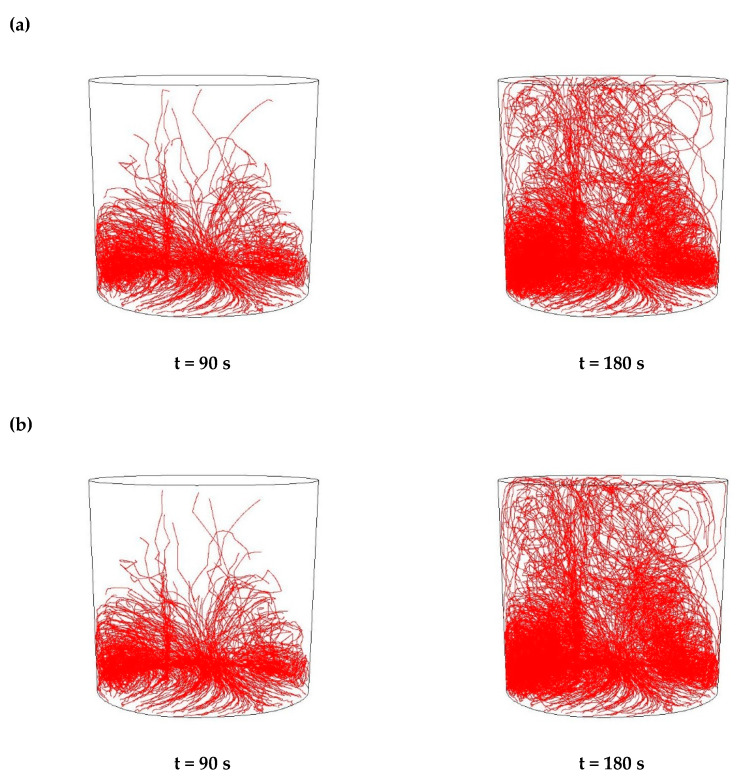
Trajectories of non-metallic inclusions MnS for the case of argon flow of 25 L/min for the particle diameters of (**a**) 18 µm and (**b**) 27 µm in the 42CrMo4 steel.

**Figure 13 materials-15-03039-f013:**
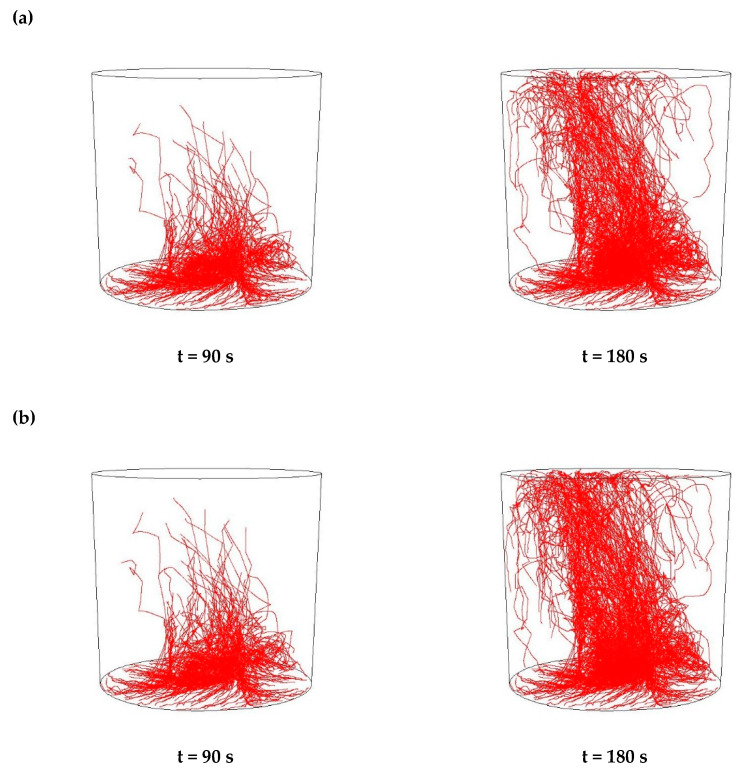
Trajectories of non-metallic Al_2_O_3_ inclusions for the case of argon flow of 50 L/min for the particle diameters of (**a**) 18 µm and (**b**) 27 µm in the 42CrMo4 steel.

**Figure 14 materials-15-03039-f014:**
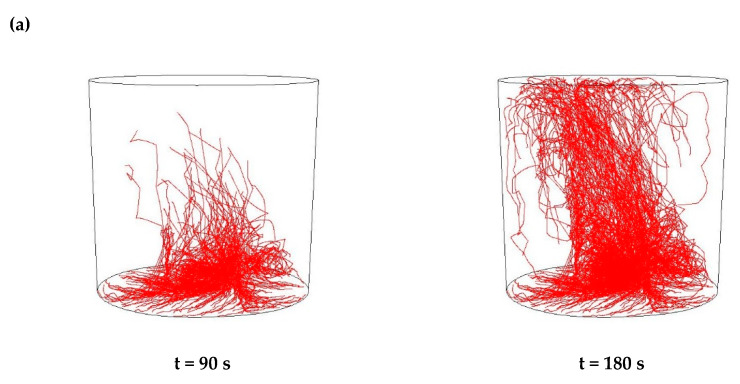
Trajectories of non-metallic inclusions MnS for the case of argon flow of 50 L/min for the particle diameters of (**a**) 18 µm and (**b**) 27 µm in the 42CrMo4 steel.

**Figure 15 materials-15-03039-f015:**
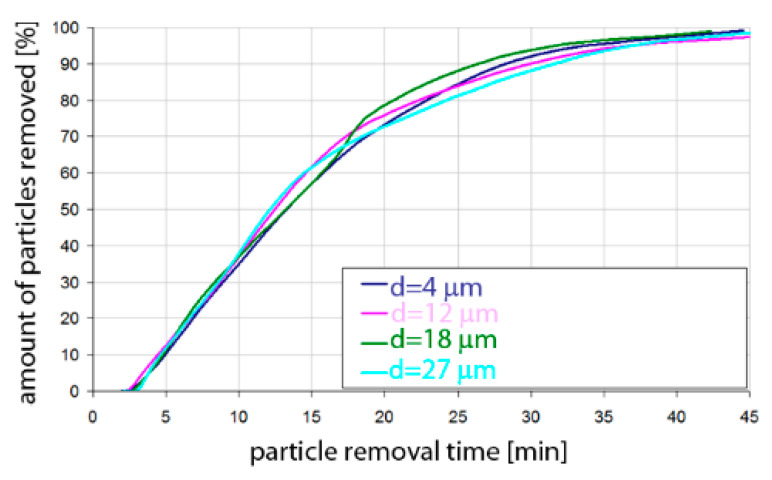
Share of removed non-metallic Al_2_O_3_ inclusions from the analysed ladle in the case of argon flow of 25 L/min through the 18CrNiMo7-6 steel.

**Figure 16 materials-15-03039-f016:**
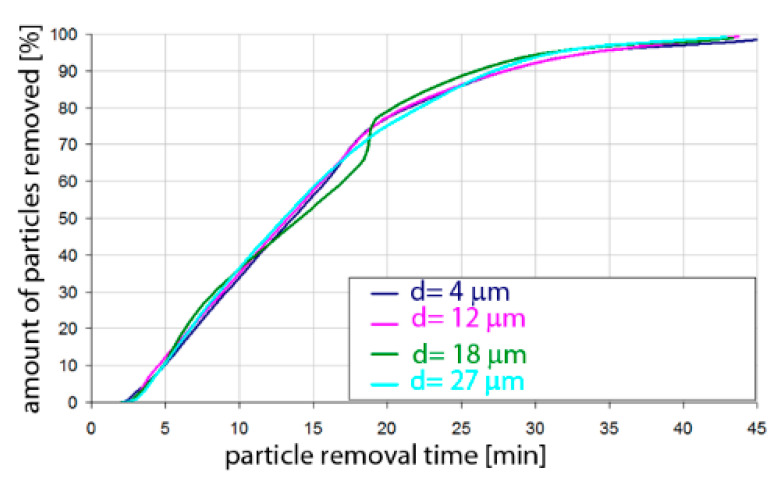
Share of removed non-metallic MnS inclusions in the analysed ladle in the case of argon flow of 25 L/min through the 18CrNiMo7-6 steel.

**Figure 17 materials-15-03039-f017:**
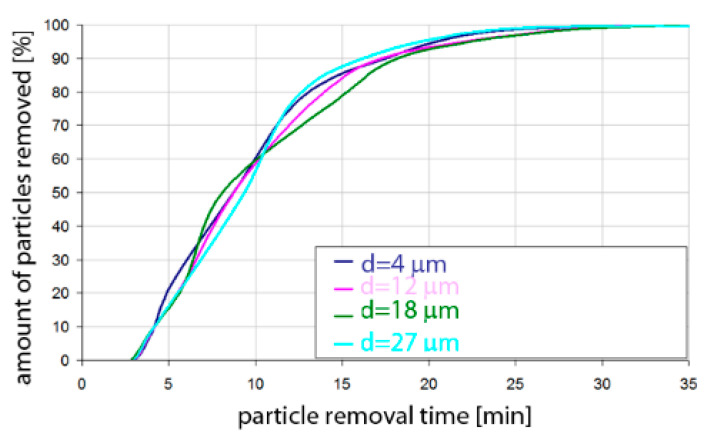
Share of removed non-metallic Al_2_O_3_ inclusions from the analysed ladle in the case of argon flow of 50 L/min through the 18CrNiMo7-6 steel.

**Figure 18 materials-15-03039-f018:**
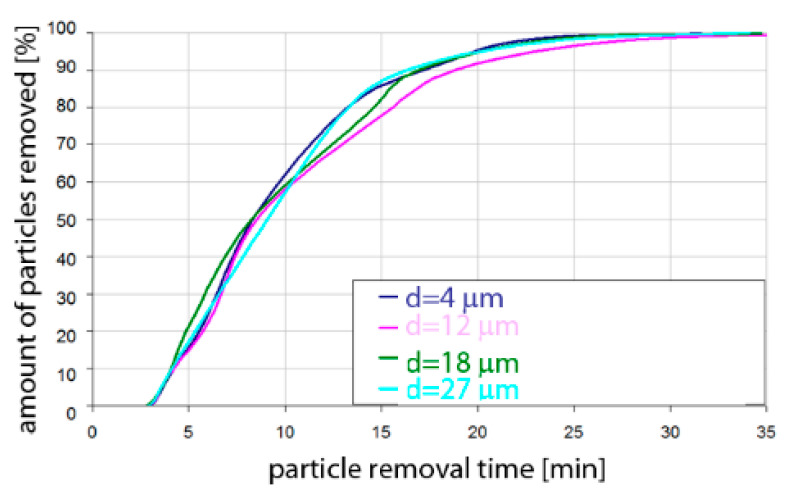
Share of removed non-metallic MnS inclusions from the analysed ladle in the case of argon flow of 50 L/min through the 18CrNiMo7-6 steel.

**Figure 19 materials-15-03039-f019:**
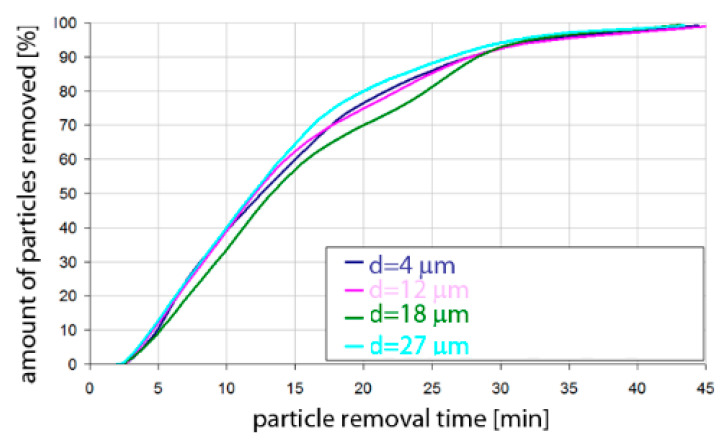
Share of removed non-metallic Al_2_O_3_ inclusions from the analysed ladle in the case of argon flow of 25 L/min through the 42CrMo4 steel.

**Figure 20 materials-15-03039-f020:**
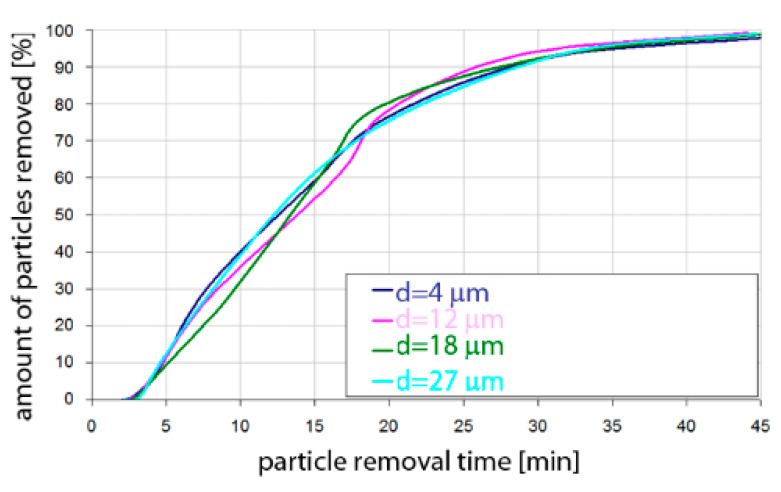
Share of removed non-metallic MnS inclusions from the analysed ladle in the case of argon flow of 25 L/min through the 42CrMo4 steel.

**Figure 21 materials-15-03039-f021:**
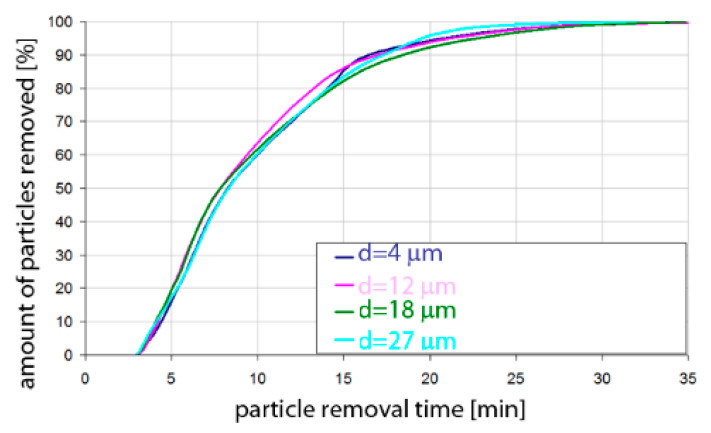
Share of removed non-metallic Al_2_O_3_ inclusions from the analysed ladle in the case of argon flow of 50 L/min through the 42CrMo4 steel.

**Figure 22 materials-15-03039-f022:**
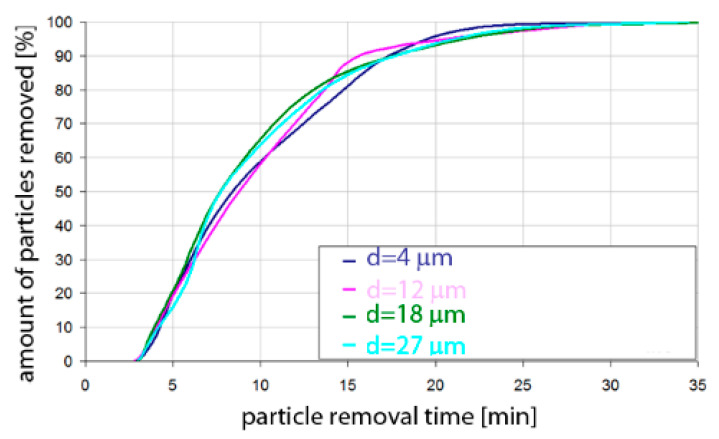
Share of removed non-metallic MnS inclusions from the analysed ladle in the case of argon flow of 50 L/min through the 42CrMo4 steel.

**Table 1 materials-15-03039-t001:** The chemical compositions of the analysed steels.

Steel Grade	C	Mn	Si	P	S	Cr	Ni	Mo
				wt%				
18CrNiMo7-6	0.18	0.65	0.24	0.010	0.008	1.65	1.45	0.26
42CrMo4	0.42	0.72	0.21	0.015	0.010	1.15	0.15	0.16

**Table 2 materials-15-03039-t002:** Calculated melting point and density of steel at various C contents.

No.	Steel Grade	C wt%	Refining Temperature °C	Density $\frac{\mathrm{kg}}{m^{3}}$
1	18CrNiMo7-6	0.18	1620	6979
2	42CrMo4	0.42	1610	7004

**Table 3 materials-15-03039-t003:** Numerical simulation variants.

Simulation No.	Steel Grade	Argon Flow L/min	Refining Temperature °C
Sim_1	18CrNiMo7-6	25	1620
Sim_2	18CrNiMo7-6	50	1620
Sim_3	42CrMo4	25	1610
Sim_4	42CrMo4	50	1610

**Table 4 materials-15-03039-t004:** Results of research on the quantitative distribution of inclusions in 18CrNiMo7-6 steel.

Sample No.	Sulphides	Oxides	Nitrides	Sulphides and Oxides <3 µm	Sum
	% of Inclusions Area				
Sample 1	0.05	0.00	0.01	0.11	0.17
Sample 2	0.16	0.11	0.00	0.24	0.51
Sample 3	0.05	1.37	0.00	0.12	1.54
Sample 4	0.28	0.00	0.04	0.17	0.49
Sample 5	0.11	0.09	0.00	0.11	0.31
Sample 6	0.23	0.00	0.06	0.18	0.47
Sample 7	0.12	0.00	0.00	0.14	0.27
Sample 8	0.29	0.32	0.00	0.18	0.79
Sample 9	0.20	0.12	0.00	0.17	0.49
Sample 10	0.04	0.19	0.00	0.08	0.30
Sample 11	0.00	0.11	0.00	0.16	0.27
Sample 12	0.00	0.31	0.17	0.24	0.72
Average	**0.13**	**0.22**	**0.02**	**0.16**	**0.53**
Standard Deviation	0.10	0.36	0.0	0.05	0.35

**Table 5 materials-15-03039-t005:** Results of research on the quantitative distribution of inclusions in 42CrMo4 steel.

Sample No.	Sulphides	Oxides	Nitrides	Sulphides and Oxides <3 µm	Sum
	% of Surface Inclusions Area				
Sample 13	0.84	0.23	0.18	0.29	1.53
Sample 14	1.45	0.08	0.03	0.83	2.39
Sample 15	1.21	0.00	0.11	0.40	1.71
Sample 16	0.93	0.42	0.11	0.46	1.92
Sample 17	0.56	0.28	0.04	0.26	1.14
Sample 18	0.57	0.49	0.03	0.33	1.43
Sample 19	0.47	0.08	0.24	0.28	1.06
Sample 20	0.88	0.46	0.08	0.42	1.84
Sample 21	0.78	0.00	0.04	0.49	1.31
Sample 22	0.56	0.22	0.06	0.18	1.01
Sample 23	1.41	0.22	0.20	0.34	2.18
Average	**0.88**	**0.22**	**0.10**	**0.39**	**1.59**
Standard Deviation	0.33	0.17	0.07	0.16	0.44

**Table 6 materials-15-03039-t006:** Considered parameters of liquid steel during refining in a ladle in the conducted numerical simulations.

Steel Grade	C wt%	Argon Flow L/min
18CrNiMo7-6	0.18	25
18CrNiMo7-6	0.18	50
42CrMo4	0.42	25
42CrMo4	0.42	50

**Table 7 materials-15-03039-t007:** The considered sizes of inclusions and their density in numerical simulations.

Non-Metallic Inclusion Type	Density of Non-Metallic Inclusions kg/m^3^	Diameter of Non-Metallic Inclusions µm
MnS	3900	4 4
Al_2_O_3_	3950	
MnS	3900	12 12
Al_2_O_3_	3950	
MnS	3900	18 18
Al_2_O_3_	3950	
MnS	3900	27 27
Al_2_O_3_	3950	

## Data Availability

Not applicable.

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
