# Peer review of "Numerical Investigation of Outflow of Non-Metallic Inclusions during Steel Refining in the Ladle"

_materials, 2022, doi:10.3390/ma15093039_

Round 1

Reviewer 1 Report

Accept in present form

Author Response

Answer for the Rewiever was written in the file.

Reviewer 2 Report

The authors addressed some of the reviewers’ comments. However, there are still some issues needed to clarify before publishing.

  1. As mentioned by the authors, the refining process and removal of non-metallic inclusions are greatly influenced by the inclusions’ composition, size, shape, and the surface phenomenon and wettability of inclusions by the liquid steel. However, the authors only considered two kinds of inclusions (Al2O3 and MnS) distinguished them by density and size in the simulations. Why the authors ignored the shape of inclusions and surface phenomenon in their simulations. How does this influence the simulation results?
  2. As shown in Fig. 3 and Fig.4, many inclusion particles are in elongated shapes instead of spherical shapes. The authors also mentioned that both spherical and elongated particles were observed in lines 238-242. Why the authors set both Al2O3 and MnS inclusions as spherical shapes with different diameters in the simulations?
  3. The images of inclusions trajectory are very similar for different inclusions, different inclusion sizes, different liquid steels. Only gas flow can influence the trajectory as shown by the authors. All the refining rate curves in Figs. 15-22 are also very similar. Is this because the simulation model used in this paper is too simplified and some important factors are ignored? As mentioned by the authors that “inclusions are probably in different physical states: Al2O3 in the solid state and MnS in the liquid state” in line 579, however, the authors did not consider this in their model. Also, the surface phenomenon of these two inclusions is probably different, but without consideration.
  4. Are there any experimental evidences to prove the simulation results?

Author Response

(The authors gave the same response as above.)

Author Response

(The authors gave the same response as above.)

Round 2

Reviewer 2 Report

The authors have addressed all the questions, and the paper can be published. The simulation result is undoubtedly of technological importance for industrial applications, however, the simulation results in some figures in the manuscript are very similar, which might be caused by the model simplicity. Since there are too many similar figures in the manuscript, it is suggested to consider whether it is possible to remove some figures to the Appendix to improve the readability of the paper.   

Reviewer 3 Report

The authors' research is extremely commendable. They presented the manuscript and revised it in response to the reviewers' comments. It is now publishable.

This manuscript is a resubmission of an earlier submission. The following is a list of the peer review reports and author responses from that submission.

Round 1

Reviewer 1 Report

The manuscript presents the results of numerical simulations of liquid steel flow in the ladle. However, there are still some problem which are needed to be solved in this manuscript.

  1. As the author said, the impurities are including oxides (Al2O3, SiO2, FeO, MnO), sulphides (FeS, MnS) and nitrides (AlN, CrN). Why the author choose Al2O3 and MnS? Is this known by the experiences? It is better that the author give some references to support this view.
  2. Actually, there are little difference between Figure 5-8,Figure 9-16 and Figure 17-24. How can a reader find some useful informations from these similar Figures?
  3. Regarding to the explanations, the differences between these Figures are over-interpreted. Meanwhile, there are too many Figures and too little explanations. As a paper, this can not be accepted.
  4. I can not see any useful way to produce high-quality steel according to the simulation of this manuscript.
  5. The results are not proved by any determinations. We can not know if the results are correct or not.

Based on these comments, this manuscript should be rejected.

Reviewer 2 Report

    This paper reported a numerical simulation study on the liquid steel flow during steel refining in the ladle. The authors investigated the effect of gas flow, steel grades, types and sizes of non-metallic inclusions on the outflow of non-metallic inclusions in the steelmaking ladle. This paper provided substantial simulation results and suggested that the gas flow had the greatest influence on the outflow of non-metallic inclusions. However, the discussion of the results should be improved. Main issues are the following:

  • In the Introduction section, the authors need to emphasize the innovation of this paper. Instead of discussing the background and software of the simulation method a lot, it is better to focus more on the factors and mechanisms influencing the outflow of non-metallic inclusions during steel refining in the ladle. And the authors need to introduce the progress of the simulation studies on this steel refining topic in the literature. What is new in this paper compared with other works in the literature?
  • Do the simulation results on the non-metallic inclusions removal rate in Figs. 17-24 agree with experiments? Can the authors present some experiment results to confirm the simulations?
  • It is suggested to move most part in the Discussion section to the Results section. The authors should describe the findings in Figs. 5-24 in the Results section. In the Discussion section, it needs to compare the similarity or differences of the findings shown in Figs. 5-24 and provide the explanations or mechanisms, instead of only describing those figures.
  • The authors claimed that “The effectiveness of the operation of removing non-metallic inclusions from the liquid steel is determined by a number of factors” in Lines 116-122 in the manuscript. However, they only investigated the effect of gas flow, steel grades, types and sizes of non-metallic inclusions on the outflow of non-metallic inclusions in the steelmaking ladle to reach the conclusion that the gas flow had the greatest influence. This simulation study ignored the reactions between the non-metallic inclusions and the liquid steel or the slag (wettability of particles by liquid steel, the ability of the slag to assimilate inclusions). Does it simplify too much in the study for non-metallic inclusions outflow?
  • What are the differences in (a) (b) (c) (d) in Figs. 9-16? It should be provided in the figure captions.

    Overall, this paper reports an interesting simulation work to guide steel refining, however, it lacks necessary discussions. Revision is needed before this paper can be published in Materials.          

Reviewer 3 Report

The article contains a number of things that would be good to clarify, refine and add before publishing.

Line 69. Apparently, in square brackets there is a reference to sources separated by a dash or a comma - [14-16].

On lines 56-62 you write that the use of specialized programs also does not give fully satisfactory calculation results. But at the same time, do not provide any supporting arguments in this paragraph. Please, make references to sources that would confirm this.

In the article it would be good to indicate why you took the steel 18CrNiMo7-642 and Cr Mo 4, why not any others? It would be good to give a justification in the article.

In my opinion, it is more convenient to read the table when the dimensions specified in the same cells as the column headers (Table 2), and not put in the table name. All the tables of the article should be done in the same style, such as Table 3.

Lines 138-142. It would be good to give the values of the selected parameters in the text or in the table.

Figures 3 and 4 – there are no scale lines.

Figures 9-16. Please, specify under the figures what canvases a, b, c, d (subplot segregation) mean.

Please, give the scheme of argon blowing of the bucket, according to which calculations were made in the article. Also, if the blowing is performed immediately after casting, then at the initial moment of time not all particles will settle to the bottom of the bucket and the initial moment of time will be different than in the results presented in Fig. 9-16. It is also written that the particles were distributed evenly along the bottom of the bucket, but it would be good to write where this assumption comes from and what it is based on. After all, if this is not the case, then the simulation results will be different.

Lines 488-489. You write that the trajectories of the particles are not affected by the grade of steel and the refining temperature. It should be clarified that this conclusion was made only for 2 grades of the studied steel. It is incorrect to make such a conclusion for all steel based on the results of modeling two similar grades.

Lines 411-415. It is stated that it is possible to select the optimal gas flow rate and the selection of the gas flow value, which gives the minimum number of dead zones. In this work, you are conducting computer modeling of processes. Please, make a number of more parameter changes and select these values and give them in this article. This will make the article more valuable from a practical point of view.

It would significantly enhance the scientific novelty of the article to provide information about the experimental verification of the results obtained.